# A Model for Reinfections and the Transition of Epidemics

**DOI:** 10.3390/v15061340

**Published:** 2023-06-08

**Authors:** Yannis C. Yortsos, Jincai Chang

**Affiliations:** USC Viterbi School of Engineering, University of Southern California, Los Angeles, CA 90089-1450, USA; jincaich@usc.edu

**Keywords:** SIR model, reinfection kinetics, delayed response, asymptotic analysis, infection wave, equilibrium state

## Abstract

Reinfections of infected individuals during a viral epidemic contribute to the continuation of the infection for longer periods of time. In an epidemic, contagion starts with an infection wave, initially growing exponentially fast until it reaches a maximum number of infections, following which it wanes towards an equilibrium state of zero infections, assuming that no new variants have emerged. If reinfections are allowed, multiple such infection waves might occur, and the asymptotic equilibrium state is one in which infection rates are not negligible. This paper analyzes such situations by expanding the traditional SIR model to include two new dimensionless parameters, *ε* and *θ*, characterizing, respectively, the kinetics of reinfection and a delay time, after which reinfection commences. We find that depending on these parameter values, three different asymptotic regimes develop. For relatively small *θ*, two of the regimes are asymptotically stable steady states, approached either monotonically, at larger *ε* (corresponding to a stable node), or as waves of exponentially decaying amplitude and constant frequency, at smaller *ε* (corresponding to a spiral). For *θ* values larger than a critical, the asymptotic state is a periodic pattern of constant frequency. However, when *ε* is sufficiently small, the asymptotic state is a wave. We delineate these regimes and analyze the dependence of the corresponding population fractions (susceptible, infected and recovered) on the two parameters *ε* and *θ* and on the reproduction number *R*_0_. The results provide insights into the evolution of contagion when reinfection and the waning of immunity are taken into consideration. A related byproduct is the finding that the conventional SIR model is singular at large times, hence the specific quantitative estimate for herd immunity it predicts will likely not materialize.

## 1. Introduction

The COVID-19 pandemic has created tremendous interest from a plethora of investigators to better understand the onset of infection waves, their propagation in time and space, and the effect of the multitude of its parameters, both physiological and behavioral. The literature abounds with such studies. Many rely on the standard SIR (Susceptible, Infected, and Recovered) model [1] and its variations. Stochastic analogs [2,3] and other extensions, e.g., those that interpret infection data using statistical techniques [4,5,6], have appeared. Here, we cited only very few examples of such studies, recognizing that many important related works have been published.

In a recent paper [7], we presented an approach to model the SIR framework by drawing analogies with chemical reaction engineering processes. Based on areal densities (namely population per unit area) rather than absolute population numbers, this approach allows for a more correct formulation characterized in terms of dimensionless parameters that express chemical and biological kinetics and also mobility and transport mechanisms (advection and diffusion). Interestingly, in the absence of spatial gradients, which is the underlying assumption of the standard SIR model, one finds that infection is governed by only one parameter, the so-called reproduction number, $R_{0}=\frac{L}{T}$, expressed in [7] in terms of physiological and behavioral properties. Here, T is the average characteristic time parametrizing infection and depends on both physiological and behavioral parameters, while L is the average characteristic time for the recovery of infected individuals (e.g., assumed 14 days, for the case of COVID-19). It was shown in [7] that in the SIR framework under the assumptions that $R_{0}$ remains constant, that no new variants emerge, and that there are no reinfections, the process asymptotically wanes towards an equilibrium state of zero infections. This state is identified as denoting the onset of herd immunity, since any new infections will not lead to a new infection wave, but to an immediate decay. In the standard SIR model this state is asymptotically approached in an exponential manner. At the same time, we also showed [8], that the approach to the state of herd immunity can be slower—for example, following a power law (rather than an exponential decay) in time, if $R_{0}$ is not constant, but increases at late times, following a power-law dependence on the infected fraction. This is indeed likely to occur, at the later stages of infection, given the societal tendency to relax behavior as the infection wanes. Under this assumption, the resulting herd immunity also differs from that predicted by the traditional SIR model of constant $R_{0}$.

The standard infection models, including those in Ref. [7], do not consider reinfection. When the latter is allowed, however, a steady state of zero infections will not be reached at large times, whether exponentially or algebraically. Instead, epidemics will evolve into different, endemic states, in which infections continue. The nature of these states, how they are reached, and how they depend on the various process parameters form the objectives of this paper. We examine whether, in the case of reinfections, there are multiple, rather than single, infection waves, what their frequency is, and what are the properties of the resulting fractions of susceptible, infected, and recovered individuals.

Reinfection has been discussed in the general literature before, using a number of different approaches. We will cite a few. Thieme and Yang [9] and Song et al. [10] proposed generic SIR models with reinfection occurring without delay. Safan et al. [11] concentrated on a two-stage model for pertussis. Ashraf [12] summarized aspects of a SIR model that involves birth rate, vaccination, and waning immunity. Okhuese [13] and McMahon and Robb [14] predicted reinfection rates for COVID-19 using a more elaborate SEIRU model. Camacho and Vargas-De-Leon [15] developed a model for disease-free equilibria, with the stability of these equilibria controlled by the reproductive number and the recurrence rate. The possibility of delay times before reinfection sets in has also been included. Finally, Cooke [16] proposed general differential-difference models for diseases, while Beretta and Takeushi [17] and Enatsu et al. [18] generalized an SIR model by also including a delay time and studied the stability of the resulting state. Mena-Lorka and Hethcote [19] used a more elaborate SIR model to describe disease equilibria but did not endeavor in reinfection processes. 

The review of the literature shows that we do not have at present a definitive approach to how to account for reinfections. Some works have proposed delay-type models, which lead to differential-difference equations. Others have postulated generic integral generation terms that lead to integrodifferential equations. This paper sheds light on this issue by adopting the reaction process-like contagion model proposed in ref. [7], here extended to include reinfections through the introduction of a novel rate of generation of new susceptible individuals. The model includes two new parameters, ε and θ, denoting respectively, the kinetics of reinfections, and a delay time for the onset of reinfections. In general, the resulting description is in terms of a single non-linear integrodifferential equation, which, however, in the limit of fast reinfection kinetics (large ε), reduces to a differential-difference equation, thus unifying the previous two different approaches in the literature. The paper extends the problem formulation, delineates the asymptotic regimes obtained, and examines the dependence of the various properties and patterns of infected, recovered, and susceptible population fractions on the three parameters, $R_{0}$, ε, and θ.

## 2. Mathematical Formulation and Results

We follow the approach in ref. [7], where contagion and recovery were represented by two equivalent chemical reactions, namely:(1)S+J→J+J
(2)J→R
where S, J and R denote susceptible, infected, and recovered individuals. To capture reinfections, we will propose an additional, equivalent chemical reaction that converts recovered populations to susceptible ones, namely:(3)R→ S
thus, leading to new susceptible individuals for infection. Reaction (3) will be assumed to be first order, with kinetics that account for various biological and behavioral factors. For the sake of generality, we will also allow the onset of the reaction to occur only after a characteristic delay in time has elapsed. Then, a recovered individual will have a finite probability of being reinfected, possibly multiple times. As in ref. [7], we will make significant simplifying assumptions, namely that there are no perished individuals, while we do not account for vaccinations that can confer additional immunity. These are made in order to focus on intrinsic, key aspects of reinfection. In short, we make two key assumptions: (1) A recovered individual can be reinfected only after a (dimensionless) time θ, has elapsed. This time is the same for each recovered individual. (2) The rate by which recovered individuals are reinfected is proportional to the recovered population fraction—namely, a recovered individual has the same probability of being reinfected as any other infected one. Additional improvements can and should be added later as desired.

We proceed in the absence of spatial gradients (corresponding to what is also known as a “batch reactor” setting [7]) and a constant reproduction number $R_{0}$. The addition of chemical reaction (3) leads to a generalized SIR model containing an additional source term. Appendix A shows that this is described by a convolution integral. This leads to the following set of dimensionless equations for the three population fractions (susceptible, infected, and recovered):(4)$$s^{\prime }\left(t\right)=-R_{0}si+\varepsilon \int_{0}^{t-\theta }i\left(\tau \right)\exp\left(-\varepsilon \left(t-\theta -\tau \right)\right)d\tau H\left(t-\theta \right)$$
(5)$$i^{\prime }\left(t\right)=R_{0}si-i$$
(6)$$\;r^{\prime }\left(t\right)=i-\varepsilon \int_{0}^{t-\theta }i\left(\tau \right)\exp\left(-\varepsilon \left(t-\theta -\tau \right)\right)d\tau H\left(t-\theta \right)$$

Here, superscript indicates time derivative, and we introduced the two new dimensionless parameters, ε and θ. In the absence of reinfections (ε=0), Equations (4)–(6) revert to the standard SIR model. Note that the source term on the RHS of (4) and (6) is a specific convolution integral, which will be discussed further below. Parameter ε is the dimensionless kinetic rate of reaction (3), describing the loss of immunity of infected individuals. It is normalized with the rate of recovery as defined in ref. [7]: namely, if I is the average characteristic time for the loss of immunity, then $\varepsilon =\frac{L}{I}$, where L was defined above as the average characteristic time for the recovery of infected individuals (e.g., assumed 14 days, for the case of COVID-19). In general, we expect small values of ε, the case of large ε corresponding to the rather unrealistic case of the instant loss of immunity for all recovered individuals (because of its significant mathematical interest, however, this case will be also explored as well). Parameter θ≥0 is a dimensionless time delay, normalized with L, after which reinfection commences. In our notation, it measures the number of time intervals, of a dimensional duration of about half a month, for COVID-19, which must elapse before reinfection is allowed. In the simulations below, we will typically focus on the range 0<θ<24 (the latter corresponding to about one-year delay). Finally, it is important also to note that Equation (9) does not include any of the two reinfection parameters.

The solution of (4)–(5) in the numerical simulations that will follow is subject to the initial conditions:(7)s(0)=0.99;i(0)=0.01; r(0)=0
while variables s, r, and i obey the closure relation:(8)s+i+r=1

We must note that in the following results, both numerical and asymptotic, the solutions of (4)–(5) satisfy the physical constraints, 0≤s≤1, 0≤r≤1 and 0≤i≤1. Proving the validity of these conditions, in general, will require elaborate mathematical proofs which will not be attempted in this paper.

We now focus on deriving the gain and loss terms in (4)–(5). Consider the set R of recovered individuals. It consists of two different subsets, $R_{1}$ and $R_{2}$. Subset $R_{1}$ contains infected individuals recovered at time τ, such that t−θ≤τ, which are not yet subject to losing immunity (see also schematic in Appendix C). Conversely, the members of set $R_{2}$ are individuals recovered at time 0<τ≤t−θ. Thus, they have started and continue losing immunity. Subset $R_{2}$ is empty if t<θ, hence the need of the Heaviside step function H(t−θ), defined as the following:$$H\left(x\right)=\{\begin{array}{c} 1,\;x>0\\ 0,\;x<0 \end{array}$$

Next, we consider a differential tranche of members in $R_{2}$, created during time Δτ. Its size is equal to i(τ)Δτ. Assuming first-order reaction kinetics for (3), this tranche loses members at the rate εi(τ)Δτ. At time t, therefore, its magnitude has been reduced to εi(τ)Δτexp(−ε(t−θ−τ)). Hence, the total loss at time t is the integral over all τ, namely $\varepsilon \int_{0}^{t-\theta }i\left(\tau \right)\exp\left(-\varepsilon \left(t-\theta -\tau \right)\right)d\tau H\left(t-\theta \right)$. We conclude that this is indeed the term on the RHS of (4) and (5).

Equations (4)–(6) will be solved subject to the typical initial conditions $i\left(0\right)=i_{0},\;s\left(0\right)\equiv s_{0}=1-i_{0},\;r\left(0\right)=0$, where $i_{0}\ll 1$ (see also (7)). Before we proceed further, we must make a couple of additional observations: First, Equations (4)–(6) are actually a system of only two differential equations, e.g., (4)–(5), expressed in terms of two variables, s and i, since variable r follows directly from the closure relation (8). In fact, one can simplify even further by solving Equation (5) for s, and substituting in (6). Then, one obtains the single non-linear, integrodifferential equation:(9)$$\frac{b}{\left(b+1\right)}{\left(\frac{i^{\prime }}{i}\right)}^{\prime }+i^{\prime }+i=\varepsilon \int_{0}^{t-\theta }i\left(\tau \right)\exp\left(-\varepsilon \left(t-\theta -\tau \right)\right)d\tau H\left(t-\theta \right)$$

Here, we defined $b\equiv \frac{1}{R_{0}-1}$, and assumed $R_{0}>1$ (which is required for the spreading of infection). Because b is the inverse of $R_{0}-1$, it is expected to vary between a value of about 0.25 (for an aggressive contagion with $R_{0}=5$) to values much larger than 1 (when contagion is weak, $R_{0}\approx 1$).

Equation (5) generalizes the SIR model to the case of reinfections. Its solution and its dependence on the various parameters will be the main focus of this paper. We must note that when ε=0, which is the SIR case, Equation (9) reverts to the standard SIR equation. This can be solved exactly with the solution [7]:(10)$$i\left(t\right)=1-r-s_{0}\exp\left(-R_{0}r\right);\;\mathrm{where}\;t=\int_{0}^{r}\frac{du}{1-u-s_{0}\exp\left(-R_{0}u\right)}$$

For future reference we will also write (10) as
(11)$$s\approx \exp\left(-R_{0}r\right)$$
where we assumed that the onset of the infection is at $s_{0}\approx 1$ (see (7)). From Equation (9) we find that in the limit corresponding to i≈0, the final value $r_{\infty }$, which solves $1-r_{\infty }-s_{0}\exp\left(-R_{0}r_{\infty }\right)=0$ is approached exponentially fast in time, with an exponent that is equal to $1-R_{0}s_{\infty }>0$.

### 2.1. Some Special Results

It is now worthwhile to first present some special results, the derivation of which is detailed in Appendix A.

i.A general expression for r(t)

Because of its linear nature, Equation (6) can be used to also provide an expression of *r*(*t*) in terms of i(t). The result is (Appendix A):(12)$$r\left(t\right)=\int_{0}^{t}i\left(\tau \right)\left\{1-H\left(t-\tau -\theta \right)\left[1-\exp(-\varepsilon \left(t-\tau -\theta \right)\right]\right\}dt$$

As expected, 0≤r(t)≤1.

ii.The *θ* = 0 limit

When θ=0, one anticipates that the loss term in (6) will reduce to εr(t), since there is no delay in time. This is indeed the case, as shown in Appendix A. Now, the relevant equations become the following:(13)$$s^{\prime }\left(t\right)=-R_{0}si+\varepsilon r\left(t\right)$$
(14)$$i^{\prime }\left(t\right)=R_{0}si-i$$
(15)$$\;r^{\prime }\left(t\right)=i-\varepsilon r\left(t\right)$$

This important limit will be explored in considerable detail. In this case, one can also use the following alternative equation to (9):(16)$$bv^{\prime \prime }-{v^{\prime }}^{2}-\left(1+2\varepsilon \right)vv^{\prime }+\varepsilon \left(\frac{1}{1+\varepsilon }+b\right)v^{\prime }+\varepsilon v\left(1-v\left(1+\varepsilon \right)\right)=0$$
where we defined the following:(17)$$v=\frac{1}{\left(1+\varepsilon \right)}-\left(b+1\right)r$$

This change of variables was motivated by the fact that v(∞)=0, thus allowing for an easier asymptotic analysis.

iii.The large *ε* limit

Even though unrealistic, it is worth examining the limit ε≫1, corresponding to fast kinetics. In this limit, and after a delay time of θ is reached, all recovered individuals are instantly susceptible to infection. Now, the gain-loss terms in Equations (5), (6) and (9) reduce to (see Appendix A):(18)$$\underset{\varepsilon \to \infty }{\lim}\varepsilon \int_{0}^{t-\theta }i\left(\tau \right)\exp\left(-\varepsilon \left(t-\theta -\tau \right)\right)d\tau H\left(t-\theta \right)=i\left(t-\theta \right)H\left(t-\theta \right)$$
and the corresponding equations read as follows:(19)$$s^{\prime }\left(t\right)\;=\;-R_{0}si+i\left(t-\theta \right)H\left(t-\theta \right)$$
(20)$$i^{\prime }\left(t\right)=R_{0}si-i$$
(21)$$\;r^{\prime }\left(t\right)=i-i\left(t-\theta \right)H\left(t-\theta \right)$$
and
(22)$$\frac{b}{\left(b+1\right)}{\left(\frac{i^{\prime }}{i}\right)}^{\prime }+i^{\prime }+i=i\left(t-\theta \right)H\left(t-\theta \right)$$

Here, we find that the integral term reduces to a difference term, and the integrodifferential equations reduce to differential-difference equations. We noted in the introduction that a number of models in biological population dynamics have used a variety of models, some based on integrodifferential equations and some on differential-difference equations. Here, we show that the latter is the limit of the integral term in the case of fast kinetics, thus unifying the two different approaches.

iv.Equilibrium states

When reinfections are present, the above formulation admits non-trivial equilibrium state solutions (denoted by subscript ∞). Assuming that these states are asymptotically stable (conditions for which will be developed below), their values are (Appendix A):(23)$$s_{\infty }=\frac{1}{R_{0}}=\frac{b}{\left(b+1\right)}$$
(24)$$i_{\infty }=\frac{\varepsilon }{\left(b+1\right)\left(1+\varepsilon \left(1+\theta \right)\right)}=\frac{\left(R_{0}-1\right)\varepsilon }{R_{0}\left(1+\varepsilon \left(1+\theta \right)\right)}$$
(25)$$r_{\infty }=\frac{\left(1+\varepsilon \theta \right)}{\left(b+1\right)\left(1+\varepsilon \left(1+\theta \right)\right)}=\frac{\left(R_{0}-1\right)\left(1+\varepsilon \theta \right)}{R_{0}\left(1+\varepsilon \left(1+\theta \right)\right)}$$

One concludes that allowing for reinfections (ε≠0) leads at large times (assuming that such states are stable) to a constant, non-zero fraction in the population of infected individuals, $i_{\infty }$, which increases as the intensity of infection ($R_{0}$) increases. Moreover, note the important result that even for infinitesimally small values of ε, the asymptotic values of $s_{\infty }$ and $r_{\infty }$ are not those predicted by the traditional SIR model (Ref. [7], and Equation (10)). The corollary is that the traditional SIR model is unstable, becoming singular at large times, because even a small value of ε will produce discontinuities in the final values $s_{\infty }$ and $r_{\infty }$. Since these values define the state of herd immunity, it is clear that the latter cannot be predicted using the SIR model and (9). We will comment on these features in more detail in the subsequent sections.

### 2.2. Numerical Results

Next, we consider the numerical solution of (4)–(6) and examine the sensitivity of the results to the values of the three parameters ε,θ, b. These define a three-dimensional space, one boundary of which, namely the plane ε=0, θ≫1, corresponds to the conventional SIR model, which is well studied. For additional useful insights, we will focus on the other two boundaries of the {ε, θ} parameter space, namely, the planes θ=0, and ε≫1. These correspond to Equations (16) and (22), respectively. We expect that they will provide prototypical regimes, valid for all other parameter values as well.

1.The case θ = 0

In the absence of a delay in time, θ=0, the problem reduces to Equations (13)–(15), which does not contain an integral source term. Numerical simulations for four different values of ε (ε= 10, 1, 0.1, and 0.01), spanning a range of behaviors, are shown in Figure 1. We observe the following: At large ε (Figure 1a) the variation of all three populations is monotonic, and after about an O(1) time interval, they asymptotically approach a stable state (which will be shown to be a stable node), which from (23) reads as follows: $s_{\infty }=\frac{1}{R_{0}}=\frac{b}{\left(b+1\right)}$, $i_{\infty }=\frac{R_{0}-1}{R_{0}}=\frac{1}{\left(b+1\right)}$, $r_{\infty }\approx \frac{1}{\left(b+1\right)\varepsilon }\ll 1$. Asymptotically, the fraction of the recovered population is small and of order $\varepsilon^{-1}$, and that there is a constant, non-zero fraction of infected individuals, that increases with decreasing b (increasing $R_{0}$), while the fraction of susceptible individuals $s_{\infty }$ decreases with increasing $R_{0}$. In this limit, closed-form results are also possible. At the large ε limit one obtains the analytical result
(26)$$i\left(t\right)\approx \frac{1}{\left(b+1\right)+\left\{\frac{1}{i_{0}}-\left(b+1\right)\right\}\exp\left(-\frac{t}{b}\right)}\;;\;r\approx \frac{i}{\varepsilon }\;;\;1-s\approx \varepsilon r$$
suggesting an asymptotic decay at the rate $b^{-1}$.

As ε decreases, the behavior changes, initially to a slight, and progressively to a stronger, undulation (Figure 1b–d). A number of distinct waves for the susceptible fraction appear, the amplitude of which decreases with time. We will show that such oscillatory behavior corresponds to a spiral (a “damped oscillator” [20]). It is worth examining Figure 1c,d in more detail. Here, over a length of time of about 2, in our dimensionless notation, the variables closely follow the SIR model of Ref. [7] (where ε=0) which is described by the SIR limit (16), namely:(27)$$bv_{0}^{\prime \prime }-v_{0}^{\prime }{}^{2}-v_{0}v_{0}^{\prime }=0$$
the integration of which gives the explicit SIR result in (10). While approximating well the early part of the curves (for example see Figure 1c), this result fails at larger times, since its asymptotic limit, reached exponentially fast, is not the correct limit $v_{\infty }=0$, but rather the solution $v_{\infty ,0}>0$ of the algebraic equation:(28)$$v_{\infty ,0}=1+b\ln\frac{z_{\infty ,0}+b}{\left(1+b\right)s_{0}}$$

As a result, and at about t=2 in Figure 1c, the number of susceptible individuals starts increasing, due to the loss of immunity of some of the recovered population, thus leading to an increase in infections, hence to the appearance of a secondary infection wave. This wave eventually wanes, but because of additional reinfections, infection continues, with the emergence of secondary waves, now of a lower amplitude, and so on. A sequence of such waves, with constant frequency, but with an amplitude that appears to decrease exponentially in time, follows. Figure 1d is a clear demonstration of this behavior.

To better analyze these results, consider the solution of (16) at large times. By neglecting higher-order terms and linearizing (namely, by conducting a linear stability analysis), we have
(29)$$v^{\prime \prime }+\left(m_{0}+\varepsilon \right)v^{\prime }+m_{0}\left(1+\varepsilon \right)v\approx 0$$
the solution of which is
(30)$$v=\exp(-\lambda_{0}t)f\left(t\right)$$

Here, f(t) satisfies
(31)$$f^{\prime \prime }+\frac{\Delta }{4}f=0$$
where we have defined
(32)$$\lambda_{0}\equiv \frac{\left(\varepsilon +m_{0}\right)}{2}>0,\;m_{0}\equiv \frac{\varepsilon }{b\left(1+\varepsilon \right)}>0,\;\Delta \left(\varepsilon ,b\right)\equiv 4m_{0}-{\left(\varepsilon -m_{0}\right)}^{2}$$

When Δ<0, namely for sufficiently large ε, such that ${\left(\varepsilon -m_{0}\right)}^{2}>4m_{0}$, the solution of (31) is a monotonic approach to a stable solution (a stable node) (Figure 1a). In the opposite case, ${\left(\varepsilon -m_{0}\right)}^{2}<4m_{0}$, it is a sinusoidal wave (a spiral) (Figure 1b–d), with frequency $\omega_{0}=\frac{1}{2}\sqrt{4m_{0}-{\left(\varepsilon -m_{0}\right)}^{2}}$. Here, within two arbitrary constants C and φ, we have
(33)$$f=C\cos\left(\omega_{0}t+\varphi \right)$$
which, when inserted in (29), provides the damped oscillator solution. The frequency $\omega_{0}$ increases with decreasing b, namely with larger $R_{0}$. It also increases with ε, until it reaches a maximum, above which it decreases with increasing ε, and vanishes when Δ=0, beyond which the asymptotic approach is no longer oscillatory.

Figure 2 shows a plot of variable f as a function of the rescaled time $\omega_{0}t$ for the latter case. The approach to the asymptotic solution is relatively fast, roughly corresponding to $t>4\lambda_{0}{}^{-1}$. For example, for ε= 1, 0.1 and 0.01, and b=0.25, it is of the order of 2.5, 17, and 160, respectively (and in dimensional notation for COVID-19 equal to 1.25 months, 8.5 months, and 6 years, respectively). The latter value is of course unrealistic, given the various other circumstances that will emerge during that period.

If we express the solution of (29) as v~exp(at), we find the characteristic equation
(34)$$a^{2}+\left(m_{0}+\varepsilon \right)a+m_{0}\left(1+\varepsilon \right)=0$$
for the generally complex number a. By further denoting a=x+iy, we find the following: (i) If Δ<0, then $x=-\lambda_{0}\pm \sqrt{-\Delta }<0$ and y=0, while (ii) if Δ>0, then $x=-\lambda_{0}<0$, and $y=\sqrt{\Delta }$. Figure 1a corresponds to the first case (Δ= −25.95), while Figure 1b–d, correspond to the second case (Δ= 7, 1.93, 0.16).

The demarcation of the region where the approach to the asymptotic state changes from that of an exponential decay to a “damped oscillator” is shown in Figure 3. The delineating curve is Δ=0, namely $\varepsilon =m_{0}-2\sqrt{m_{0}}$. (There is also another branch where Δ=0, but which is not shown in Figure 3 because it corresponds to the physically unrealistic results of very large $R_{0}$).

We conclude that in the absence of a delay in time, two stable asymptotic regimes emerge when reinfections are allowed, both leading to an asymptotically constant state. The traditional SIR model cannot capture the long-time behavior, even when ε is infinitesimally small. Indeed, while the SIR model is followed for small times, (e.g., less than about 2 in Figure 1c), a different regime sets in when time is larger, leading to a different asymptotic state. The conclusion is that the SIR model is singular at times of order $\varepsilon^{-1}$.

For future use it is useful to also plot the results in the phase plane (s, r). Figure 4 shows results for the case b=1 ($R_{0}=2)$. Plotted in the figure are also two limiting curves, one corresponding to the SIR model (Equation (11)) and another corresponding to the case where s(t) reaches an extremum (maximum or minimum). This can be readily shown to be the curve $r=R_{0}\frac{s\left(1-s\right)}{\varepsilon +sR_{0}}$. When the solution trajectory intersects this curve, its derivative vanishes and its direction changes. The transition from a stable node to a spiral, namely to a wave behavior, as the value of ε decreases is clear in Figure 4.

2.The limit of large *ε*

Consider, next, a different boundary in the parameter space, the plane defined by ε≫1. This is the case where reinfection occurs instantly, after a delay time θ has elapsed. Figure 5 shows numerical results corresponding to four different values of θ, selected such that $\theta <\theta_{m}$, $\theta_{m}<\theta <\theta_{c}$, and $\theta >\theta_{c}$, respectively, where the two critical values $\theta_{m}$ and $\theta_{c}$ depend on b (and for Figure 5, they are equal to $\theta_{m}\left(1\right)=0.96$ and $\theta_{c}\left(1\right)=7.8$, respectively). We observe the following:

In Figure 5a,b, where, $\theta <\theta_{m}$ and $\theta <\theta_{c}$, respectively, the solution approaches at the large ε limit the equilibrium state (23)–(25)
(35)$$s_{\infty }=\frac{1}{R_{0}}=\frac{b}{\left(b+1\right)}$$
(36)$$i_{\infty }=\frac{1}{\left(b+1\right)\left(1+\theta \right)}=\frac{\left(R_{0}-1\right)}{R_{0}\left(1+\theta \right)}$$
(37)$$r_{\infty }=\frac{\theta }{\left(b+1\right)\left(1+\theta \right)}=\frac{\left(R_{0}-1\right)\theta }{R_{0}\left(1+\theta \right)}$$

The approach is different depending on the value of θ. When $\theta <\theta_{m}$ (Figure 5a), there is a monotonic exponential decay (corresponding to a stable node); when $\theta_{m}<\theta <\theta_{c}$ (Figure 5b), the solutions have the features of a damped oscillator (spiral); and when $\theta >\theta_{c}$ (Figure 5c,d), the asymptotic equilibrium is no longer stable, and the variables are approaching a periodic structure pattern of constant amplitude, with a period that depends on θ and b. This transition is consistent with a Hopf bifurcation [21], expected to arise when the rate of growth of the amplitude at large times becomes positive, and suggests the possibility of a limit cycle behavior. We note that in the case of large θ, there are periods during which the infected fraction is infinitesimally small. We believe that this characteristic of any reinfection process, since after a time θ has elapsed, recovered individuals can become susceptible leading the system away from conditions of herd immunity, hence the onset of new infection waves. Of course, all this is under the assumption that all physiological and biological processes remain the same (no vaccinations, no changes in the biology of the recovered individuals, etc.).

To quantify the results shown in Figure 5, we consider the linear stability of Equation (22) around the equilibrium state (36). Assuming a small perturbation ϖ of i around $i_{\infty }$ and linearizing at large times we find
(38)$$\frac{1}{m_{\infty }}\varpi^{\prime \prime }+\varpi^{\prime }+\varpi =\varpi \left(t-\theta \right)\;$$
where we defined
(39)$$m\equiv \frac{\varepsilon }{b\left(1+\varepsilon \left(1+\theta \right)\right)}\;\mathrm{and}\;m_{\infty }\equiv \underset{\varepsilon \to \infty }{\lim}m\left(\theta ,\varepsilon \right)=\frac{1}{b\left(1+\theta \right)}$$

The solution of (38) (which is the counterpart of (29) in the ε≫1 limit) is the exponential
(40)ϖ~exp(at)
where a is a complex number. Substituting in (38) we find
(41)$$\frac{1}{m_{\infty }}a^{2}+a+1-\exp\left(-a\theta \right)=0$$
and by further taking a=x+iy we obtain the system of the two algebraic equations
(42)$$\frac{1}{m_{\infty }}(x^{2}-y^{2})+1+x=\exp\left(-x\theta \right)\cos\left(y\theta \right)$$
(43)$$\frac{2}{m_{\infty }}xy+y=-\exp\left(-x\theta \right)\sin\left(y\theta \right)$$
where without loss, we can take y>0. Equations (42) and (43) do not have a simple explicit solution. However, one can show (Appendix B) that there exist two critical values $\theta_{m}\left(b\right)$ and $\theta_{c}\left(b\right)$ delineating three different regimes: (i) a stable node, if $0<\theta <\theta_{m}\left(b\right)$; (ii) a spiral, if $\theta_{m}\left(b\right)<\theta <\theta_{c}\left(b\right)$; and, (iii) a periodic state, arising from a Hopf bifurcation, if $\theta >\theta_{c}\left(b\right)$. As expected, the limit θ=0 in (42)–(43) coincides with the corresponding limit of (30)–(31) in the case of large ε. For future use, it is also worth presenting the results in the phase plane (s, r) (Figure 6), which shows clearly the three asymptotic states. This figure will be used in the discussion section that follows.

For completeness we note the analytical result in the case of small θ (see Appendix B)
(44)$$i\left(t\right)=\frac{1}{\left(b+1\right)\left(1+\theta \right)+\left\{\frac{1}{i_{0}}-\left(b+1\right)\left(1+\theta \right)\right\}\exp\left(-\frac{t}{b}\right)}\;;\;s\left(t\right)\approx 1-\left(1+\theta \right)i\left(t\right)\;;\;s\left(t\right)\approx 1-\frac{\left(1+\theta \right)}{\theta }r\left(t\right)$$
suggesting an asymptotic decay at the rate $b^{-1}$. Equation (44) reproduces the numerical result of Figure 5a, while it is also consistent with the θ=0 case discussed previously at the large ε limit.

Figure 7 shows the dependence of $\theta_{m}\left(b\right)$ and $\theta_{c}\left(b\right)$ on the parameter b (hence on the reproduction number $R_{0}$), obtained from the solution of (42)–(43) (Appendix B). As the intensity of the epidemic increases (larger $R_{0}$), it is easier for the asymptotic state to become a periodic pattern of constant amplitude and frequency. We anticipate somewhat similar behavior for finite or smaller ε, which is the next case to be discussed.

3.The general case

Having explored the three limiting boundaries in the (ε, θ) parameter space, we can now consider the more general problem, where ε is finite and θ is non-zero. We expect that, in general, there will emerge three different regimes: Two stable asymptotic states, for sufficiently small values of θ, the nature of which could be either a stable node or a spiral, and an unstable equilibrium state at sufficiently large $\theta >\theta_{c}\left(b,\varepsilon \right)$, which will lead to an oscillatory periodic state (a limit cycle). Numerical results are shown in Figure 8 and Figure 9 for a fixed b, for different values of θ, and for two different values of ε, respectively. In Figure 8 ε is chosen so that it falls in the upper region of Figure 3 (where Δ<0) where the asymptotic state for θ=0 is a stable node. Figure 9, on the other hand, corresponds to a smaller value of ε, chosen to be in the lower region in Figure 2, (where Δ>0) where the asymptotic state for θ=0 is a spiral.

Figure 8a,b (θ=0.5 and 5) suggest that the solution approaches a stable equilibrium, monotonically in Figure 8a, and as a damped oscillator in Figure 8b. The other figure, Figure 8c,d (where θ= 10 and 20) indicate that the equilibrium state is unstable and becomes a periodic solution. This is consistent with the limit of large ε, discussed before, and where the order of appearance of the asymptotic regimes as θ increases from zero were monotonic, a damped oscillator or a periodic state, respectively. Two critical values $\theta_{m}\left(b,\varepsilon \right)$ and $\theta_{c}\left(b,\varepsilon \right)$ separate these asymptotic regimes. 

On the other hand, Figure 9 shows that there are only two regimes, a damped oscillator, for sufficiently small θ (Figure 9a) and a periodic solution, at larger θ (Figure 9b). Again, a critical value $\theta_{c}\left(b,\varepsilon \right)$ separates these two regimes. Here, the sequence of the asymptotic regimes as θ increases from zero are a damped oscillator and a periodic state, respectively.

To confirm these results, we conducted a linear stability analysis. Appendix C shows that now the rate of growth a satisfies the algebraic equation:(45)$$\frac{1}{m}a^{2}+a+1-\frac{\varepsilon }{a+\varepsilon }\exp\left(-a\theta \right)=0$$

As expected, this reduces to the corresponding limits, namely Equation (32) for the case of θ=0, and Equation (41) for the case of large ε, respectively, With the usual representation a=x+iy, we further find (Appendix C).
(46)$$\left(x^{2}-y^{2}+mx+m\right)\left(x+\varepsilon \right)-\left(2x+m\right)y^{2}-\varepsilon m\exp\left(-x\theta \right)\cos\left(y\theta \right)=0$$
(47)$$\left[\left(x^{2}-y^{2}+mx+m\right)y+\left(2x+m\right)y\left(x+\varepsilon \right)\right]+\varepsilon m\exp\left(-x\theta \right)\sin\left(y\theta \right)=0$$

The solution of (46) and (47) is similar to the previous case of large ε (Figure 7) only if ε is larger than a ctitical value, $\varepsilon >\varepsilon_{c}\left(b\right)$, at which point $\theta_{m}=0$. For such values of ε, there are two critical values, $\theta_{m}\left(\varepsilon ,b\right)$ and $\theta_{c}\left(\varepsilon ,b\right)$, that demarcate the three regions where there is monotonic exponential decay, a damped oscillator or a constant oscillation (Figure 10). However, when ε is smaller than the critical value, $\varepsilon <\varepsilon_{c}\left(b\right)$, (such that Δ>0), a critical value $\theta_{m}>0$ does not exist, and there is only one critical value $\theta_{c}\left(\varepsilon ,b\right)$ that separates a regime of a damped oscillator from that of a constant oscillation. The value $\varepsilon_{c}\left(b\right)$ is the same as the one plotted in Figure 3, since it demarcates the region at which Δ=0 and θ=0. As shown in Figure 3, $\varepsilon_{c}\left(b\right)$ decreases as b increases or $R_{0}$ decreases. Finally, it is also not difficult to show that the asymptotic behavior of $\theta_{c}$ when ε≪1 is given by the expression
(48)$$\theta_{c}\approx \frac{\sqrt{2\left(b+1\right)}}{\sqrt{\varepsilon }}$$
which is consistent with Figure 10b. For completeness, the results of Figure 8 are also plotted in the phase space plane (s, r) (Figure 11). The three asymptotic states are clearly portrayed. Similar phase portraits can also be obtained for the simulations shown in Figure 9.

## 3. Discussion

To provide a physical interpretation of the results obtained, it is useful to analyze the behavior in the phase space (s, r), as portrayed in Figure 4, Figure 6, and Figure 11. In all cases, the solution trajectories lie within the right triangle bounded by the two coordinates s, r (which are varying in (0,1)). As expected from the closure relation (8), the dotted green line (the hypotenuse) is the case of zero infections, and any line parallel to it denotes a fixed value of i. In the Figures, the color red denotes the trajectory corresponding to the SIR model (where ε=θ=0), given by Equation (11). For a better understanding of the results, it is worth focusing first on Figure 4, where θ=0, and subsequently on Figure 6, where ε≫1.

Consider, first, Figure 4. In all these cases the equilibrium state cannot be one in the absence of infections, (namely $i_{\infty }=0$). Otherwise, any recovered individual will become instantly susceptible, thus leading to an increase in s, at which point infection will recommence. Instead, the equilibrium state must correspond to $s_{\infty }=\frac{1}{R_{0}}$ (where the time derivative of i also vanishes). In the first two panels (Figure 4a,b), where the asymptotic solution is a stable node, the solution trajectory is different from that of the SIR model and approaches the different equilibrium state $s_{\infty }=\frac{1}{R_{0}}=0.5$. In the limit of large ε, the solution is a straight line, following Equation (26), as approximately shown in Figure 4a. Because of the absence of time delay and because ε is sufficiently large, reinfection is instant and fast, thus starting to interfere with the SIR process immediately. Thus, the infection process is reinforced from its onset, as new supplies of susceptible individuals are provided fast from those just recovered. Hence, one expects a fast and exponentially asymptotic approach to the equilibrium point.

In Figure 4a,b the infected fraction increases monotonically (as also shown in Equation (26)), while the susceptible fraction decreases monotonically as it approaches $s_{\infty }$. At equilibrium, there is a constant and fast conversion of recovered populations to infected ones, and vice versa, with infection and reinfection rates balancing each other, such that the system is at an equilibrium. Any deviation from equilibrium is rapidly restored. The characteristic time for this approach decreases as $R_{0}-1$ increases. This dependence on $R_{0}$ is notable, as the latter incorporates social behavior aspects, such as spatial distancing, face coverings, etc., increasing as the latter decrease.

When the asymptotic attractor is a spiral (Figure 4c,d), where ε is relatively small, infection first follows the SIR behavior described in (11). Because of reinfection, however, the rate of decrease of the susceptible population slows down, as some of the recovered individuals lose immunity to join the susceptible population. The maximum in the fraction of infected individuals occurs when s approaches the value $s_{\infty }=\frac{1}{R_{0}}$ (see Equation (5)), which in the phase plane is a horizontal line parallel to the r-axis. This endows the solution trajectory with a counter-clockwise motion, since the second derivative of i at that point is negative, as can be readily shown by taking the derivative of (5) and evaluating at $s_{\infty }=\frac{1}{R_{0}}$. As time proceeds, and because of reinfection, the trajectory will intersect at some time the curve $r=R_{0}\frac{s\left(1-s\right)}{\varepsilon +sR_{0}}$, shown in light blue in Figure 4. This will signal that s has reached a minimum (the RHS of (13) vanishes at that point). The counter-clockwise motion continues with decreasing infection populations, until s increases to the value $s_{\infty }=\frac{1}{R_{0}}$, where the fraction of infected individuals now reaches a minimum, following which infection rates increase again. At a sufficiently large value of s the rates of infection exceed the rates of replenishment, due to reinfections, the trajectory intersects again the curve $r=R_{0}\frac{s\left(1-s\right)}{\varepsilon +sR_{0}}$, but at a different point, and a new infection wave sets in. The difference with the first wave is that because of the smaller overall susceptible fraction (which is close to $\frac{1}{R_{0}}$), the infection rate is correspondingly smaller, hence the trajectory is closer to the equilibrium state, which is given by $s_{\infty }=\frac{b}{\left(b+1\right)}$, $i_{\infty }=\frac{\varepsilon }{\left(b+1\right)\left(1+\varepsilon \right)}$, $r_{\infty }=\frac{1}{\left(b+1\right)\left(1+\varepsilon \right)}$. Eventually, the system starts spiralling towards this state, which it approaches with a decreasing amplitude. At equilibrium, there will be progressively smaller infection waves, driven by the conversion of those recovered to susceptible, which are progressively attenuated due to the relatively low rates of infection and the fact that close to the equilibrium state $s=\frac{1}{R_{0}}$, the net rate of generation of infected individuals is zero. In Figure 6 and Figure 11, the stable node and spiral behaviors are very similar to Figure 4. In Figure 6a,b, the value of the delay time θ is relatively small and thus interferes relatively fast with the SIR model dynamics, thus the trajectory starts deviating from that of the SIR model relatively soon. A similar behavior occurs in Figure 11a,b.

When the delay time θ is sufficiently large, for example as in Figure 6c,d, the trajectory does not depart from the SIR model until it almost reaches its asymptotic limit. When t begins exceeding θ, Equation (21) becomes the following:(49)$$\;r^{\prime }\left(t\right)=i-i\left(t-\theta \right)$$

Following that time, the infection fraction i will continue decreasing. The trajectory approaches the hypotenuse, while i(t−θ) will continue increasing, due to the time lag resulting from the delay time. At some time, the RHS of (49) will vanish, and the trajectory will commence reversing its direction ($\;r^{\prime }\left(t\right)<0$), consistent with the onset of reinfections. As long as $R_{0}s<1$, the fraction of infected individuals will keep decreasing (see Equation (5)), implying an even closer approach of the trajectory to the hypotenuse, until s reaches the value $\frac{1}{R_{0}}$, at which point the trajectory will start curving away from it. Because of the corresponding increase in i, as time increases, there will be a time at which the RHS of (49) will become positive again, thus the trajectory will start curving towards increasing values of r, and the cycle will recommence. Thus, the periodic trajectory obtained at sufficiently large values of the delay time θ consists of the following: one segment, almost coinciding with the SIR curve, and another segment being very close to the hypotenuse, where the infection fraction is negligible (Figure 6d and Figure 10d). The characteristics of the latter segment can be readily obtained by using Equations (19) and (21) and by taking i≪1. Then,
(50)$$s^{\prime }\left(t\right)\approx i\left(t-\theta \right)$$
(51)$$\;r^{\prime }\left(t\right)\approx -i\left(t-\theta \right)$$

Hence,
(52)$$s\left(t\right)\approx 1-r\left(t\right)\approx \int_{\theta }^{t}i\left(\tau -\theta \right)d\tau +s_{\infty ,SIR}=\int_{0}^{t-\theta }i\left(\tau \right)d\tau +s_{\infty ,SIR}$$
where, the SIR limit, $s_{\infty ,SIR}$ satisfies Equation (10) in the limit i→0, namely:(53)$$s_{\infty ,SIR}=\exp\left(-R_{0}\left(1-s_{\infty ,SIR}\right)\right)$$

Using the expressions for the SIR limit in (52), we find
(54)$$s\left(t\right)\approx r_{SIR}\left(t-\theta \right)+s_{\infty ,SIR}$$
where $r_{SIR}\left(t\right)$ is obtained by (10), namely:(55)$$t=\int_{0}^{r_{SIR}}\frac{du}{1-u-s_{0}\exp\left(-R_{0}u\right)}$$

At large times we have,
(56)$$s\left(t\right)\to r_{\infty ,SIR}+s_{\infty ,SIR}=1\;$$

The corresponding pattern has a period equal to θ.

In the general cases depicted in Figure 8 and Figure 9, the behavior can be explained following a similar reasoning (see also Figure 11). The analysis is more complex at smaller values of θ, and for more general values of ε, since analytical results are not easily obtained. It is important to note, however, that when ε is sufficiently small, $\varepsilon <\varepsilon_{c}\left(b\right)$, which as Figure 3 shows is likely to be the case for most realistic situations, the predominant pattern is a spiral. For a periodic pattern to emerge requires large delay times, to exceed $\theta_{c}$, as also shown in Equation (48) and Figure 10. One concludes that in the general case, where ε is not large, the asymptotic behavior is likely to be a wave with a decaying amplitude and constant frequency, approaching asymptotically a constant state.

## 4. Conclusions

In this paper we considered the extension of the traditional SIR model, as further refined in ref. [7], to capture the possibility of reinfections. We proceeded by assuming that infected individuals are reinfected at a specified kinetic rate, and that they do not acquire permanent immunity. While this is a likely unrealistic assumption it does provide an interesting long-time behavior. By appropriately capturing the rate of reinfection in terms of a convolution integral, we examined the nature of the asymptotic state as a function of three dimensionless parameters—the normalized reinfection kinetic rate ε, expected to be a small positive number; a delay time θ before which immunity is lost and reinfection commences; and the inverse of the distance of the reproduction number from unity, $b=\frac{1}{\left(R_{0}-1\right)}$, which appears to be the appropriate relevant parameter. We found that there are three possible asymptotic states. For sufficiently small θ, the process approaches a stable attractor, either in the form of a stable node, at sufficiently large kinetic rates ε, or in the form of a spiral, when ε is sufficiently small. This wave behavior is encountered at larger b (namely at milder forms of epidemics). Such behavior will set in regardless of the value of the kinetic constant of reinfection, no matter how small that might be. A wave behavior will set in when ε is not large, assuming that θ is not too small. This is likely to be the practical case. Then, the asymptotic behavior will be in the form of waves of decreasing amplitude and in the approach to a steady state, in which the fraction of infected individuals is a non-zero constant. We explored its amplitude and frequency. When the delay time for the onset of the loss of immunity is sufficiently large, the asymptotic state is not stable, and the process will settle in a periodic pattern, which consists of a segment that parallels the standard SIR model, and of another segment, where the infection fraction is negligible. This pattern has a period proportional to θ.

The model we used is based on a number of assumptions, some of which might be directly relaxed, and some that require significant additional work. One such assumption is that all recovered individuals are subject to reinfection. Clearly, this is not accurate, as fraction r also includes individuals who perish. This can be rectified if we assume that a constant fraction p of the population r perishes. Then, the above remains valid subject to the substitution ε→pε. Under the assumptions considered, it captures a model of the transition of a pandemic to an endemic state. Importantly, we find that the herd immunity values predicted by the traditional SIR model are unlikely to materialize, since even in the presence of a small reinfection rate, the system will approach the value $1-s_{\infty }=1-\frac{1}{R_{0}}$, rather than the value in Equation (31) implied in the standard SIR model. The former value is also predicted when the large-time decay is algebraic (as in [8]).

We should also note that the present model assumes that conditions exist such that the simpler SIR model (in the absence of reinfections) can be applicable, which implies compartmental areas with negligible spatial gradients. If such conditions do not exist, for example, if mobility and diffusion are important parameters, then one could use the general model presented in reference [7] and expand models (3)–(6) to also include spatial gradients due to diffusion or other mobility processes. Of course, in any data comparisons, one would also need to account for vaccinations, etc., which were not included in our paper. This can be undertaken in further studies.

## Figures and Tables

**Figure 1 viruses-15-01340-f001:**
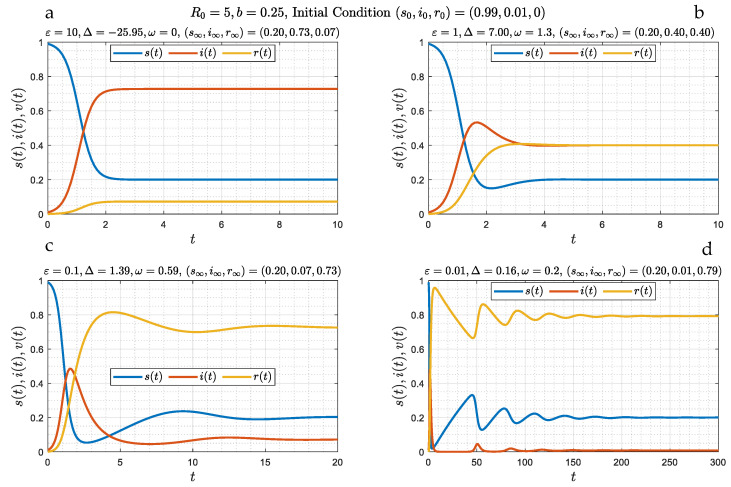
(**a**–**d**): Numerical simulations of the evolution of susceptible, infected and recovered fractions as a function of time for the case of zero delay (*θ* = 0) and four different values of *ε* (10, 1, 0.1 and 0.01). The initial conditions for the simulations were defined in (7). The asymptotic state corresponds to either a stable node (panel (**a**)), or a spiral (panels (**b**–**d**)). It is reached in a monotonic way at large *ε* (panel (**a**)), and through waves of a decaying amplitude (panels (**b**–**d**)) depending on whether parameter Δ, defined in (31), is negative (panel (**a**)) or positive (panels (**b**–**d**)), respectively.

**Figure 2 viruses-15-01340-f002:**
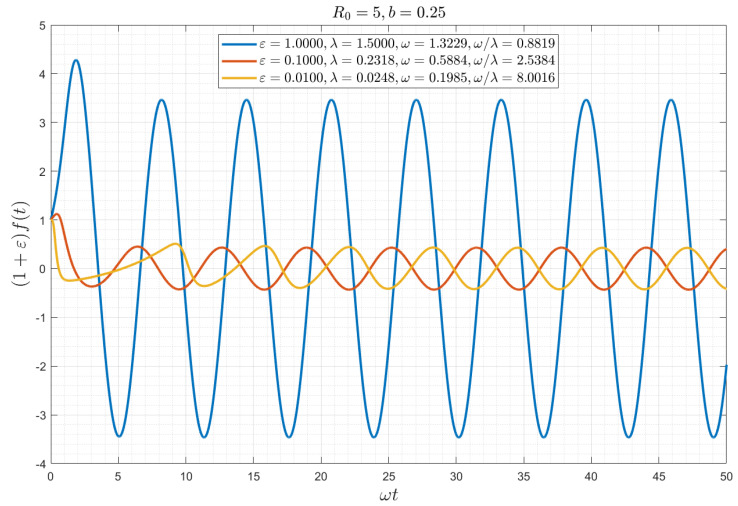
Numerical solution of Equation (16), for the zero-delay case *θ* = 0, plotted as a function of the rescaled time *ω*_0_*t* for different values of *ε* and for *b* = 0.25 (*R*_0_ = 5). Note the fast approach to the asymptotic state of a sinusoidal wave.

**Figure 3 viruses-15-01340-f003:**
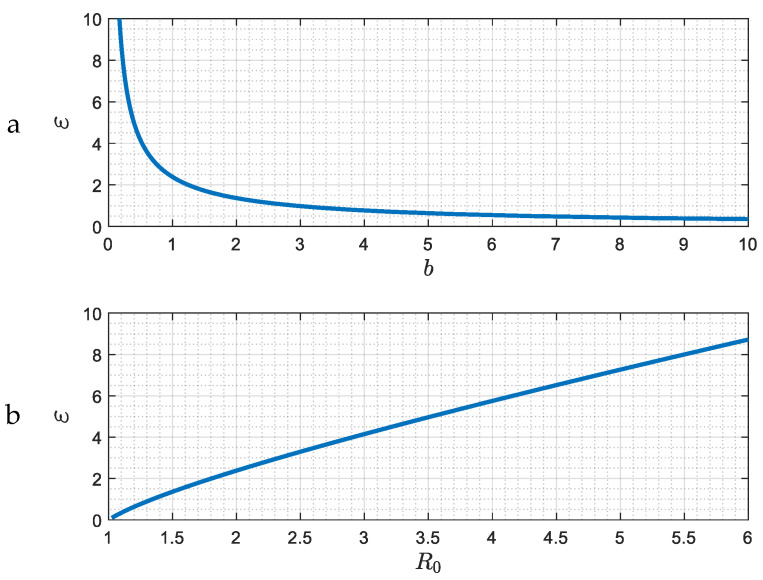
(**a**,**b**) The demarcation in the parameter space *ε*, *b* (or *R*_0_) of the two different asymptotic behaviors. Regions above or below the curve correspond to exponential decay (Δ < 0), or a damped oscillator, (Δ > 0).

**Figure 4 viruses-15-01340-f004:**
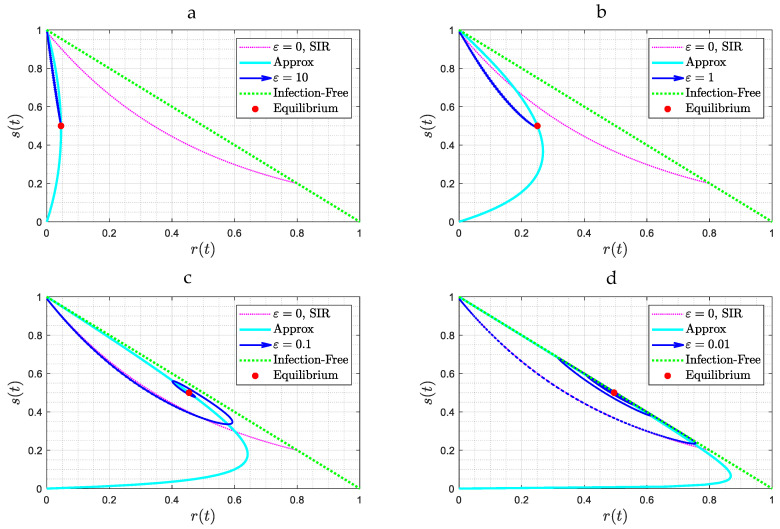
(**a**–**d**): Solution trajectories in the parameter space (*s*, *t*) for the case *θ* = 0, for four different values of *ε* and for *b* = 1. Note the attraction to a stable node (Figure 4a,b) or to a spiral (Figure 4c,d). In red is the curve corresponding to the SIR model, while in light blue is the curve at the intersection of which the trajectory reverses direction.

**Figure 5 viruses-15-01340-f005:**
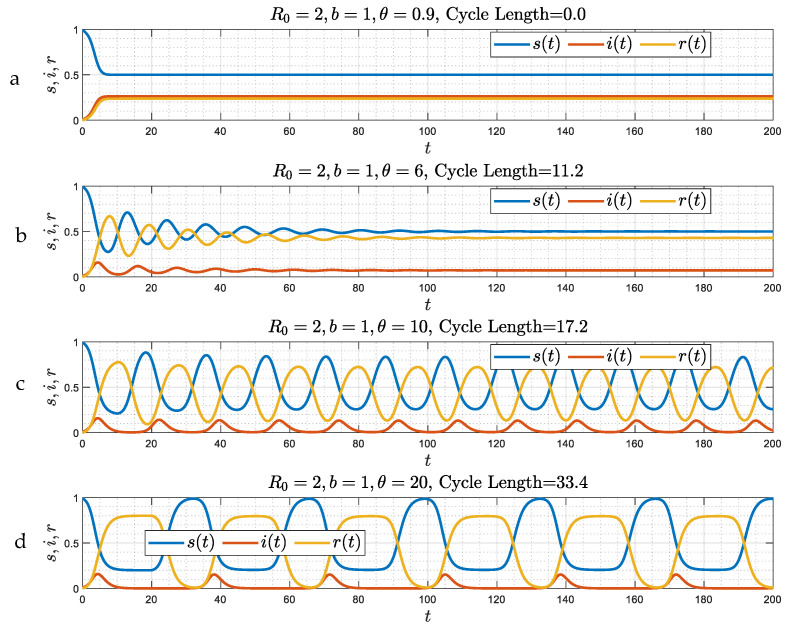
(**a**–**d**) The numerical solution of Equations (19)–(22), where ε≫1, for *b* = 1 and for four different values of *θ*, corresponding to $\theta <\theta_{m}\left(1\right)$, $\theta_{m}\left(1\right)<\theta <\theta_{c}\left(1\right)$ (panels (**a**,**b**)), and $\theta >\theta_{c}\left(1\right)$ (panels (**c**,**d**)), where $\theta_{m}\left(1\right)=0.96$ and $\theta_{c}\left(1\right)=7.8$. The asymptotic behavior is either monotonic (stable node) (Figure 5a), a damped oscillator (spiral) (Figure 5b), or an oscillation of constant amplitude (Figure 5c,d).

**Figure 6 viruses-15-01340-f006:**
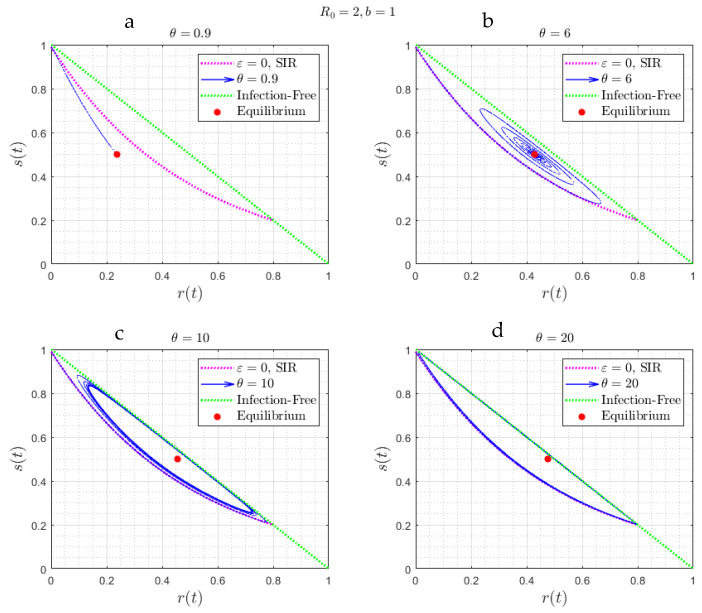
(**a**–**d**) The trajectories of the solution of Figure 5, in the parameter space (s, r). Note the attraction to a stable node (Figure 6, corresponding to $\theta <\theta_{m}\left(1\right)=0.96$), a spiral (Figure 6b, corresponding to $\theta_{m}\left(1\right)<\theta <\theta_{c}\left(1\right)=7.8$) and a limit cycle (Figure 6c,d, corresponding to $\theta_{m}\left(1\right)<\theta <\theta_{c}\left(1\right)$). In red is the curve corresponding to the SIR model (Equation (11)).

**Figure 7 viruses-15-01340-f007:**
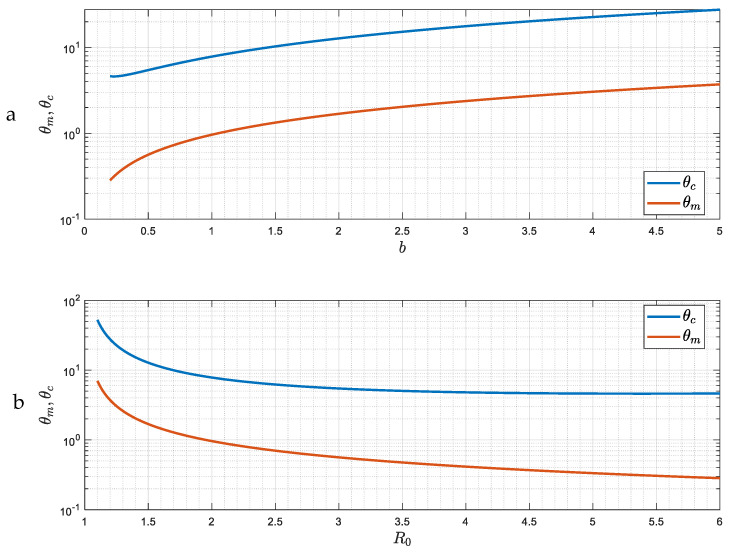
(**a**,**b**) The dependence of the two critical delay time values *θ_m_* and *θ_c_* on *b* (and hence on the reproduction number *R*_0_) for the case of large *ε*.

**Figure 8 viruses-15-01340-f008:**
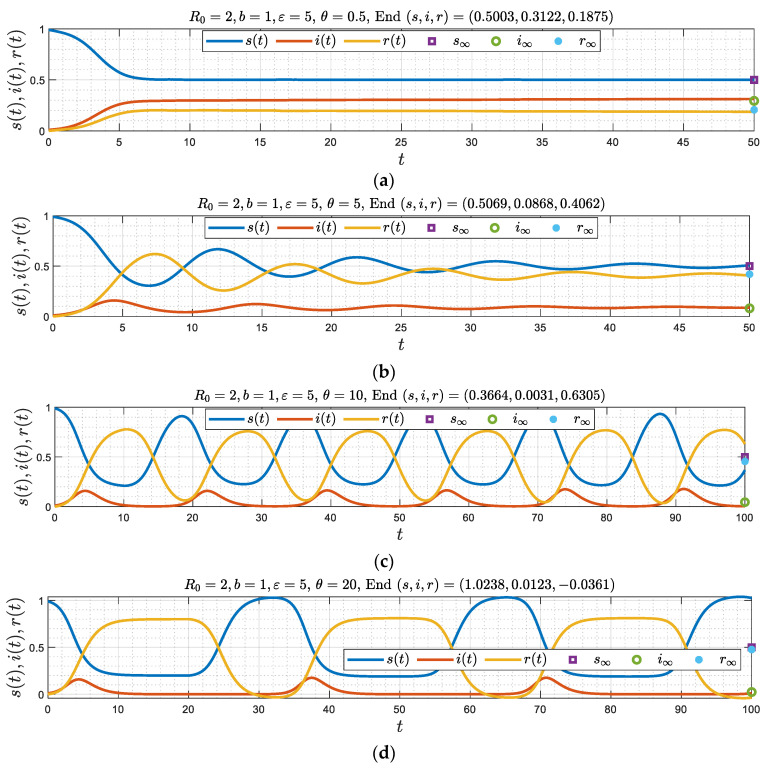
(**a**–**d**) Numerical simulations corresponding to b=1 and ε=5, where Δ<0, for four different values of θ. Note the emergence of three different regimes as θ increases ($\theta_{m}\left(1\right)=0.58,\;\theta_{c}\left(1\right)=7.8$).

**Figure 9 viruses-15-01340-f009:**
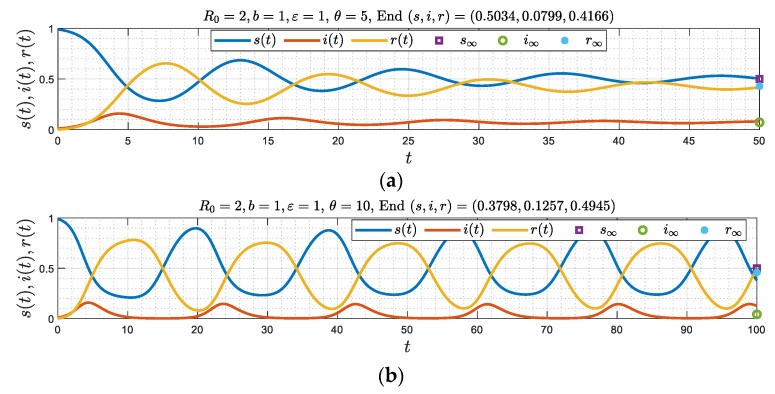
(**a**–**c**) Numerical simulations corresponding to b=1 and $\varepsilon =1,\;\theta_{c}\left(1\right)=7.4$, where Δ>0, for three different values of *θ* = 5, 10 and 20. Two different regimes emerge, a damped oscillator (spiral) in Figure 9a and a periodic pattern in Figure 9b,c.

**Figure 10 viruses-15-01340-f010:**
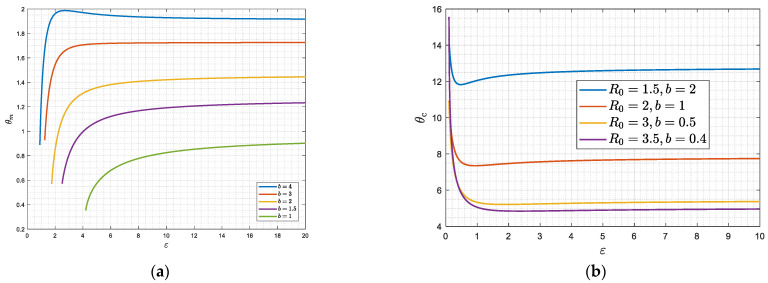
(**a**,**b**): The dependence of the critical delay times $\theta_{m}$ and $\theta_{c}$ on ε for different values of b (hence the reproduction number $R_{0}$). Note that there is no value of $\theta_{m}$ if $\varepsilon <\varepsilon_{c}\left(b\right)$.

**Figure 11 viruses-15-01340-f011:**
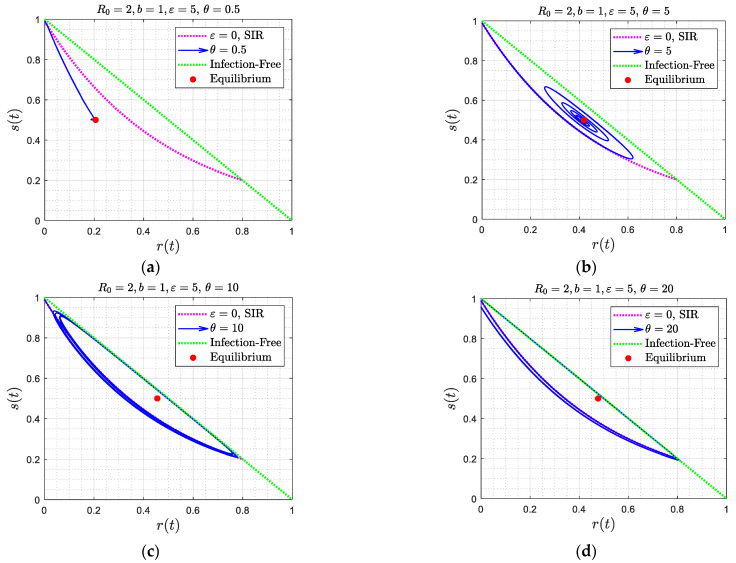
(**a**–**d**): The trajectories of the solution of Figure 7 in the parameter space (s, t). Note the attraction to a stable node (Figure 7a, where $\theta <\theta_{m}$), to a spiral (Figure 7b, where $\theta_{m}<\theta <\theta_{c}$) and to a limit cycle (Figure 7c,d) where $\theta_{m}<\theta <\theta_{c}$). In red is the curve corresponding to the SIR model.

## Data Availability

Not applicable.

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
