# Peer review of "A Model for Reinfections and the Transition of Epidemics"

_viruses, 2023, doi:10.3390/v15061340_

Round 1

Reviewer 1 Report

I think that the topic to be considered by authors seems interesting in biology.

The authors considered the factor of reinfections of infected individuals in an SIR integral differential model (1)-(3) with finite time delay. 

Then, detailed numerical simulations are given to show the variations of dynamic properties of the model (1)-(3) with respect to two parameters $\theta$ (time delay) and $\epsilon$ (the infection rate).

For example, it is observed that the solution curves become oscillated for larger time delay $\theta$.

In mathematics, I think that the following basic problem should be also considered.

1. What is the initial condition for the model (1)-(3)? Usually, to determine the solution of the model (1)-(3), the corresponding initial condition is needed.

2. While the initial condition is given, how to ensure the solution $(s(t),i(t),r(t))$ is positive or nonnegative for all $t\geq 0$? Usually, detailed proofs are needed.

3. Simple analysis on the existence of the equilibria of the model (1)-(3) are also needed, as well as their stability.

4. I want to know that whether the model (1)-(3) can be used to the data predictions of COVID-19 for some districts or countries in the world.

5. The information of the references need to be carefully checked in details.

Author Response

Response to Reviewer #1

We thank the reviewer for finding the paper of interest. Regarding the specific comments made, the following action has been taken:

  1. The initial conditions for the solution of equations (1)-(3) are the following

; ;

This is now indicated in the revised version.

  1. The reviewer asks the very astute question of how one can ensure that the solution of (1)-(3) for variables , , and leads to solutions that are positive (and we might also add, to solutions for and  that are in the interval [0,1]). At this point, the answer we can provide is that all simulations, as well as the asymptotic analyses we conducted in the various limits, lead to results with the expected behavior, namely ,  , . We agree that elaborate theoretical proofs will be needed to support the numerical and asymptotic evidence we find. This would be an interesting study for mathematicians, but it will not be attempted here.  A related comment is added in the revised text.
  2. The reviewer comments that a simple analysis of the existence of the equilibria of the model (1)-(3) are also needed, as well as their stability. This is indeed done in the original manuscript (e.g., equilibria are described in iv. Equilibrium States, while their stability is described in equations (17), (21), and (25), as well as in the text and figures that follow the respective equations).
  3. The reviewer raises the important question of how can (1)-(3) be used in predictions of COVID-19 for various districts or countries. There are two answers to this question: a. If conditions exist such that the simpler (in the absence of re-infections) SIR model is applicable, which implies compartmental areas with negligible spatial gradients, then our present model applies equally as well. b. However, if such conditions do not exist, for example, if mobility and diffusion are important parameters, then one could incorporate this model into the general model presented in reference H. Ramaswamy, A. A. Oberai and Y.C. Yortsos, A Comprehensive Spatial-Temporal Infection Model of Chemical Engineering Science, (2020), to also include spatial gradients due to diffusion or other mobility processes. Of course, in realistic data comparisons, one would also need to account for additional processes, such as vaccinations, etc., not included in our paper. Again, this can be undertaken in further studies. We have added these comments in the discussion section.
  4. We reviewed again the references and ensured that they correspond to the proper position in the cited text. The changes made are indicated in blue.

Reviewer 2 Report

In the article “A Model for Reinfections and the Transition of Epidemics” the authors investigate the dynamics of a reasonably straightforward SIRS epidemic model with delayed waning immunity. A thorough analytical analysis is presented that generates many insightful approximations, and this is accompanied by comprehensive numerical simulations. The mathematics appears correct, however no code has been provided to validate the numerical results. Overall, I found the paper to be interesting and believe it would make a valuable contribution to the literature. However, before I can recommend it for publication, I do think the current presentation could benefit from some restructuring, along with several other minor comments outlined below.

Major:

·         I felt that the current presentation, at times, makes the analysis more difficult to follow than is necessary. To begin, several equations that appear in the introduction could be moved to the second sections on “Mathematical Formulation and Results”. Moreover, the appearance of equations (1)-(3) seemed relatively abrupt, and the paper could perhaps benefit from a softer introduction to this main system (such as that presented in lines 140-152). There are many instances where the authors give vague references to earlier or subsequent results: i.e., “see above” or “see below”, that appear several pages removed from the current discussion. It is possible that some rearrangement may help alleviate these issues.

·         The authors make the interesting point that the “SIR model is singular at large times, hence the specific state of herd immunity it predicts will not materialize”. From what I understand this observation primarily stems from the endemic equilibrium solutions derived from the SIRS model which do not vanish in the limit of vanishing reinfection. From the discussion (e.g., L46-48), I think that the authors might be confusing the herd immunity threshold (the number of immune individuals required to prevent subsequent epidemic waves) with the final size (the number of people who ultimately become infected).

·         Related to the point above, the authors might like to comment on the realism of their model, and in particular the recurrent waves observed in e.g., Figure 5, where the infected population appears to become vanishingly small between cycles.

·         Please provide a link to the code used to generate all numerical results and figures presented in the article. Without this, it is impossible to verify the results.

Minor:

·         The authors might consider citing the related article: Mena-Lorcat, J., Hethcote, H.W. Dynamic models of infectious diseases as regulators of population sizes. J. Math. Biol. 30, 693–716 (1992). https://doi.org/10.1007/BF00173264

·         The authors should be consistent with the use of COVID or COVID-19.

·         Equations should be properly punctuated (e.g., the equation on L147 should be followed by a period.)

·         Panels within figures should be labelled (e.g., (a), (b)), particularly if they are referred to within the captions and main text.

·         Is the SIR reference curve in each phase portrait red or pink?

·         L18: Possible missing word after “critical”, e.g., value, threshold?

·         L255: “and” should be “an”

·         L289: I suspect there is an error in equation (13b) as the variable  does not seem to be defined.

·         L778: Forgive my ignorance, but it is not immediately obvious to me why the right- and left-hand sides of (B3) should be tangential to one another at the root of (B3). Could you please elaborate on this?

Author Response

Response to Reviewer #2

We would like to thank the reviewer for his generally positive comments about the paper. We appreciate the many useful comments made, the action to which is as indicated below.

  1. The first paragraph summarizes the suggestions made in the subsequent paragraphs, which are addressed below.
  2. The referee suggests that several equations that appear in the introduction could be moved to the second section on “Mathematical Formulation and Results”. This is now implemented in the revised version. The referee also suggests that the paper “could benefit from a softer introduction to this main system (such as that presented in lines 140-152)”. This is now implemented. Finally, the referee suggests that “there are many instances where the authors give vague references to earlier or subsequent results: i.e., “see above” or “see below,” that appear several pages removed from the current discussion. It is possible that some rearrangement may help alleviate these issues.” We revised the paper in various places to remove such ambiguities. All these significant edits are denoted in blue in the revised manuscript.
  3. Regarding the comment on herd immunity and the fact that the SIR model is singular at late times, we mention the following, which are hopefully now clearer in the revised manuscript: The standard SIR model predicts that at large times, the asymptotic solution for the fraction of susceptible and recovered individuals will be and , respectively, where solves the algebraic equation . Because at that limit any small changes in the infected population do not lead to a new epidemic (these will decay exponentially fast), the fraction  resulting from the solution of this equation is the herd immunity for that value of . However, if we allow for re-infection, no matter how small, the asymptotic limit changes to equation (12c), namely . For example, if we take , we now find  , which differs from the previous result except in the trivial case . We hope that this clarifies our point.
  4. The reviewer asks to provide a comment regarding the realism of the recurrent waves in the case of Figure 5, corresponding to large , given that there are periods during which the infected fraction is negligible. We believe that this is characteristic of any re-infection process, since after a time has elapsed, recovered individuals can become susceptible leading the system away from conditions of herd immunity, hence to the onset of new infection waves. Of course, all this is under the assumption that all physiological and biological processes remain the same (no vaccinations, no changes in the biology of the recovered individuals, etc.). This is now inserted in the revised text.   
  5. Regarding the comment on the numerical code, please note that all the numerical simulation codes were written in MATLAB, and the ODE solutions heavily relied on the built-in functions ODE45, ODE23. An iterative procedure was used to solve the integro-differential equations. The nonlinear equations were mostly solved with MATLAB function fzero(), fsolve().  3D-plot functions such surf, surfc were used in helping defining the initial guesses for the solutions in Appendices B and C. The MATLAB codes will be provided.      
  6. We now cite the paper by Mena-Lorka, J. and Hethcote, H.W., Dynamic Models of Infectious Diseases as Regulators of Population Sizes, J. Math. Biol., Vol. 30 (1992) 693-716.
  7. Assorted remarks and action taken:
  • The authors should be consistent with the use of COVID or COVID-19: COVID-19 is now consistently used.
  • Equations should be properly punctuated (e.g., the equation on L147 should be followed by a period.): Done.
  • Panels within figures should be labelled (e.g., (a), (b)), particularly if they are referred to within the captions and main text: The figures sub-graphs are now re-labeled to include (a), (b).
  • Is the SIR reference curve in each phase portrait red or pink?: It’s pink or magenta.  The descriptions are now corrected.
  • L18: Possible missing word after “critical”, e.g., value, threshold?: It is value.
  • L255: “and” should be “an”: Corrected.
  • L289: I suspect there is an error in equation (13b) as the variable does not seem to be defined: Variable was inherited from previous revisions of this paper.  The equation is now corrected.
  • L778: Forgive my ignorance, but it is not immediately obvious to me why the right- and left-hand sides of (B3) should be tangential to one another at the root of (B3). Could you please elaborate on this: Equation (B3) has a real root as long as the two curves defining the left-hand side and the right-hand side, respectively, intersect. The limiting condition for this to occur is when the two curves become tangent with one another at the intersection point. Beyond this limit, the two curves do not intersect any longer. We added this explanation in the text.

Reviewer 3 Report

In this work, the authors extend the usual SIR model cosidering loss of immunity of recovered individuals by including two new dimensionless parameters, $\epsilon$ and $\theta$, corresponding to the kinetics of reinfection and a delay time, after which re-infection
starts.    First of all, I am not sure that Viruses is the best journal for such a paper. I am afraid that this work is too mathematical for the general readership of the journal. A technical comment: the equations appear in a rather strange way in the manuscript, which makes it a bit hard to read it. Another minor comment: using braces for references is unusual. Of course, this does not affect the values of the paper.    The introduction of the parameters L, I, \epsilon, \theta is a bit confusing.  I am wondering how much the way immunity loss appears in this model contributes to the model being more realistic than other ways seen in earlier works (e.g. just writing -$\epsilon$*r(t) in the r equation).    It should be made clear which results are only based on the numerical simulations and which are actually shown analytically. More details should be provided in the Appendices. What do you mean by "equilibrium states at large times"? How about "Assuming that these states are asymptotically stable [...]  their values are" (l. 229)? How do the expressions for the equilibria depend on stability? Stability should not be "assumed".   A couple of typos: line 240: corrolary line 255: and->and

Author Response

Response to Reviewer #3

We thank the reviewer for the comments and suggestions. We address in italics the various points below:

  1. In this work, the authors extend the usual SIR model considering loss of immunity of recovered individuals by including two new dimensionless parameters, $\epsilon$ and $\theta$, corresponding to the kinetics of reinfection and a delay time, after which re-infection starts. First of all, I am not sure that Viruses is the best journal for such a paper. I am afraid that this work is too mathematical for the general readership of the journal.

We believe that this comment is best directed to the editor.  

  1. A technical comment: the equations appear in a rather strange way in the manuscript, which makes it a bit hard to read it.

Our original manuscript was submitted as a word document. We believe that upon its conversion to the journal format the equations ended up in a strange way. The current version has corrected this deficiency.

  1. Another minor comment: using braces for references is unusual. Of course, this does not affect the values of the paper.

We now use the journal’s format.  

  1. The introduction of the parameters L, I, \epsilon, \theta is a bit confusing. I am wondering how much the way immunity loss appears in this model contributes to the model being more realistic than other ways seen in earlier works (e.g. just writing -$\epsilon$*r(t) in the r equation).

We make two key assumptions: 1. A recovered individual can be re-infected only after a (dimensionless) time , has elapsed. This time is the same for each recovered individual.  2. The rate by which recovered individuals are re-infected is proportional to the recovered population fraction- namely, a recovered individual has the same probability of being re-infected as any other re-infected one. This is now inserted in the revised version.

  1. It should be made clear which results are only based on the numerical simulations and which are actually shown analytically. More details should be provided in the Appendices.

The only analytical results possible for the entire range of times are with the original SIR model (see reference 7). However, we can derive analytical results in various limits and particularly at the asymptotic limit of large times. This has been indicated in the main text (e.g. in equations (12), (17)-(19), (2), (25)-(26), (29)).

  1. What do you mean by "equilibrium states at large times"? How about "Assuming that these states are asymptotically stable [...] their values are" (l. 229)? How do the expressions for the equilibria depend on stability? Stability should not be "assumed".

It is mentioned in the text that as time increases, the system can approach one of three states: (1) a stable node (namely the three variables approach in a monotonic way an equilibrium value, which is stable); (2) a stable spiral (namely the three variables approach in an oscillatory way (waves of decreasing amplitude) an equilibrium value, which is stable); (3) a limit cycle (namely the three variables approach an unstable equilibrium value, hence the asymptotic limit at large times is a periodic pattern that continues indefinitely). We have addressed the values of these equilibrium states in the section iv. Equilibrium states and their stability in equations (17), (19) and (25). No assumption is made on their stability, which is derived following a systematic analysis (leading sometimes to a stable equilibria and sometimes to an unstable equilibria and a limit cycle). We hope that these address the reviewer’s concerns. 

  1. A couple of typos: line 240: corrolary line 255: and->and

Typos corrected.

Round 2

Reviewer 3 Report

I think that the authors have clarified the questions raised by the referees.